# Protocols Targeting Afferent Pathways via Neuromuscular Electrical Stimulation for the Plantar Flexors: A Systematic Review

**DOI:** 10.3390/s23042347

**Published:** 2023-02-20

**Authors:** Anastasia Papavasileiou, Anthi Xenofondos, Stéphane Baudry, Thomas Lapole, Ioannis G. Amiridis, Dimitrios Metaxiotis, Themistoklis Tsatalas, Dimitrios A. Patikas

**Affiliations:** 1Laboratory of Neuromechanics, School of Physical Education and Sports Science at Serres, Aristotle University of Thessaloniki, 62110 Serres, Greece; 2Physical Education and Sports Sciences, Frederick University, 1036 Nicosia, Cyprus; 3Laboratory of Applied Biology, Research Unit in Applied Neurophysiology (LABNeuro), Université Libre de Bruxelles, 1070 Brussels, Belgium; 4Université Jean Monnet Saint-Etienne, Lyon 1, Université Savoie Mont-Blanc, Laboratoire Interuniversitaire de Biologie de la Motricité, F-42023 Saint-Etienne, France; 52nd Orthopaedic Department, Papageorgiou Hospital, 56429 Thessaloniki, Greece; 6Department of Physical Education and Sport Science, University of Thessaly, 42100 Trikala, Greece

**Keywords:** neuromuscular electrical stimulation, triceps surae, plantar flexors, tibial nerve, human, stimulation parameters, afferent feedback, extra force, fatigue, training

## Abstract

This systematic review documents the protocol characteristics of studies that used neuromuscular electrical stimulation protocols (NMES) on the plantar flexors [through triceps surae (TS) or tibial nerve (TN) stimulation] to stimulate afferent pathways. The review was conducted according to the Preferred Reporting Items for Systematic Reviews and Meta-analyses (PRISMA) statement, was registered to PROSPERO (ID: CRD42022345194) and was funded by the Greek General Secretariat for Research and Technology (ERA-NET NEURON JTC 2020). Included were original research articles on healthy adults, with NMES interventions applied on TN or TS or both. Four databases (Cochrane Library, PubMed, Scopus, and Web of Science) were systematically searched, in addition to a manual search using the citations of included studies. Quality assessment was conducted on 32 eligible studies by estimating the risk of bias with the checklist of the Effective Public Health Practice Project Quality Assessment Tool. Eighty-seven protocols were analyzed, with descriptive statistics. Compared to TS, TN stimulation has been reported in a wider range of frequencies (5–100, vs. 20–200 Hz) and normalization methods for the contraction intensity. The pulse duration ranged from 0.2 to 1 ms for both TS and TN protocols. It is concluded that with increasing popularity of NMES protocols in intervention and rehabilitation, future studies may use a wider range of stimulation attributes, to stimulate motor neurons via afferent pathways, but, on the other hand, additional studies may explore new protocols, targeting for more optimal effectiveness. Furthermore, future studies should consider methodological issues, such as stimulation efficacy (e.g., positioning over the motor point) and reporting of level of discomfort during the application of NMES protocols to reduce the inherent variability of the results.

## 1. Introduction

Neuromuscular electrical stimulation (NMES) is a technique that involves involuntary muscle activation via flow of electrical current, and is commonly used to enhance and/or restore neuromuscular function [1]. More often, the electrical stimulation is delivered over the muscle, recruiting mainly the motor units underneath the surface of the electrodes [2] in a non-physiological order, i.e., synchronously and spatially fixed. In an effort to mimic the motor unit recruitment order observed during a voluntary contraction [3], further NMES protocols have been developed to activate synaptically motoneurons by stimulating afferents fibers from muscle spindles [4], and thereby recruiting from small to large motoneurons [5]. These protocols have special stimulation attributes, which have been explored with various study designs in order to document the effectiveness on stimulating afferent pathways with NMES.

Conventional NMES consists of short duration trains of pulses (50–400 μs) over the muscle belly and targets mainly the efferent pathway, by stimulating the motor axons beneath the stimulation electrodes [2]. On the other hand, NMES delivered over the muscle belly or nerve trunk with a wide-pulse duration (≥0.5 ms) [6,7], high frequency (≥80 Hz) and low stimulation intensity is referred as Wide Pulse High Frequency NMES (WPHF), and has the potential to preferentially activate motor neurons via spinal pathways [8,9,10]. This is because WPHF depolarize more effectively sensory than motor axons, as sensory axons have greater diameter, longer strength-duration time constant (chronaxie) and lower rheobase than motor axons [9,11].

Muscle and nerve stimulation that involve sensory pathways may potentiate spinal excitability and reinforce activity-dependent plasticity in neural circuits, especially when WPHF is combined with voluntary contractions [5,12]. Besides neural adaptations at spinal level, WPHF NMES may also lead to some supraspinal adaptations [13], which could be important for basic functional tasks, such as gait and balance, and further raise the interest in the investigation of NMES protocols on the lower limbs [14]. For instance, WPHF NMES-induced afferent volleys may activate motor and sensory cortical areas of the brain, as proved with functional magnetic resonance scans [15]. On the other hand, there are studies that failed to observe neuromuscular adaptations after intervention with WPHF, and dictate the necessity to standardize and optimize the stimulation properties, such as frequency, pulse duration and intensity [16].

The extensive use of NMES has consequently resulted in a wide range of protocols. By the strict definition of WPHF there is no distinction in the stimulation characteristics when stimulating over the muscle belly or the nerve trunk. Nevertheless, there are NMES protocols with shorter pulse width and lower stimulation frequencies that demonstrated similar effectiveness stimulating the afferent pathway, as typical WPHF protocols do (e.g., 11,15), but this issue is not well documented. This could be important for NMES protocols applied over nerves since the discomfort level increases with higher stimulation frequency and pulse duration. Examining the afferent pathway with nerve and muscle NMES may improve our picture about neuromuscular functions, because the stimulation site may involve neural or muscular mechanisms to a different extent [7,17,18,19,20]. For instance, nerve WPHF protocols generate fatigue patterns similar to voluntary contractions for the same force output [21]. However, later studies suggest that WPHF over the muscle or nerve trunk can generate similar adaptations, at least regarding the force development [7].

Considering the functional importance of the plantar flexor muscles during gait and balance, and the contribution of afferent pathways to sensory-motor integration during these tasks [22,23,24], documentation of the diversity of NMES protocols that stimulate the afferent pathway could be the first step to define the relevance of stimulation properties relative to the objective of NMES applications. Therefore, this review aims to document and describe the attributes of NMES protocols that give evidence of stimulating the motor neurons synaptically, via afferent pathways and have been applied on the triceps surae muscle (TS) or the tibial nerve (TN), in healthy adults. Knowing which are the NMES characteristics that are currently used to investigate underlying mechanisms of neuromuscular functions, can give insights into how NMES could be applied in the future.

## 2. Methods

The structure of this review was designed following the Preferred Reporting Items for Systematic Reviews and Meta-Analysis (PRISMA) guidelines [25].

### 2.1. Eligibility Criteria

The selected studies in this review were cohort, cohort analytic or randomized controlled trial studies that met the following inclusion criteria: (1) original research articles, (2) including interventions with NMES applied on TN or TS or both (3) of healthy adults. Studies with at least one WPHF protocol were included. WPHF protocols were protocols with stimulation frequency greater or equal to 80 Hz, rectangular pulse, and pulse duration greater or equal than 0.5 ms [8,26]. Any other protocol that had the same effect on the main outcome variable as the WPHF protocol was included for analysis as well. No other criterion was set, to allow a more comprehensive overview of this nascent field.

### 2.2. Information Sources and Search Strategy

The development of our search strategy was based on guidance from previous relevant papers [27,28]. A systematic search was conducted in The Cochrane Library, PubMed, Scopus, and Web of Science from inception to 26 July 2022. Additionally, the citations from the included studies were searched for additional studies. The complete search strategy is presented in Appendix A.

### 2.3. Study Collection and Selection Process

The collection and selection process of this review is presented in detail in Figure 1. A total number of 796 records were identified from four databases. After filtering online each database for language (English), type of publication (article, final report), and studies on humans, 151 studies were excluded and the remaining 645 were saved in a spreadsheet. All records, with title and abstract, were screened and 191 duplicate entries were removed. The titles of the remaining studies (*n* = 448) were identified as “not relevant” or “potentially relevant” by two reviewers (AP, TT), using a semi-automated process, with specific keywords search, to exclude animal studies (e.g., mice, cats), studies with not healthy population (e.g., multiple sclerosis, spinal cord injury, cerebral palsy) while keeping in consideration all inclusion and exclusion criteria. The potentially relevant studies (*n* = 91) sought for retrieval and 85 abstracts were retrieved. All abstracts were screened for eligibility by two independent reviewers (AP, AX) and the reasoning for exclusion was recorded. After this process, the reviewers compared their results and decided which articles should finally be included in the review. Five additional records were identified through citation sources from the retrieved records. All discrepancies in decisions between the reviewers were resolved through consensus. A third author (DM) acted as an arbiter when needed. After the final inclusion (*n* = 32), two authors (AP, AX) processed independently the full-text articles to extract all data items.

### 2.4. Data Extraction Process

The data items that were extracted from the manuscripts were the characteristics of the participants (number of participants and dropouts, gender, age), stimulation type (muscle or nerve) and NMES attributes (pulse width, frequency, duration, intensity). The number of dropouts due to pain was recorded, if reported. Discomfort rate was expressed as the percentage of dropout cases due to discomfort from the initial number of participants of each study. The extracted data of the two reviewers were compared to confirm their accuracy and full agreement was achieved.

### 2.5. Quality Assessment

Two reviewers assessed independently the risk of bias of the included studies (AP, AX) using the checklist of the Effective Public Health Practice Project Quality Assessment Tool (EPHPP). Disagreements were resolved by consensus with a third author acting as an arbiter (TT), when necessary.

### 2.6. Data Synthesis

For the data synthesis, protocol attributes were extracted from the studies and were summarized for each of the stimulation types (TS and TN). The scope of each study was presented narratively, and each study was classified into 4 major categories depending on the field of interest when using WPHF. Mechanisms: when studies aimed at elucidating neuromuscular mechanisms associated with WPHF. Extra forces: when WPHF was applied to generate additional force through central pathways. Fatigue: when WPHF was used as a tool to investigate muscle fatigue. Training: when WPHF was used as an intervention method. The outcome variables from the studies such as number of participants, gender, age, dropouts, pulse width, frequency patterns, duration, and intensity were extracted and categorized in custom ranges. Outcome measures were expressed as a percentage of the total number of protocols on each type of NMES.

## 3. Results

### 3.1. Study Selection

Thirty-two studies met the rigid inclusion criteria of this review and were included in the final analysis. Nineteen studies involved NMES on the TS, 10 on the TN and 3 on both. Regarding the study design, 6 were randomized controlled trials [16,29,30,31,32,33], two were cohort analysis studies [34,35] and 24 were cohort studies [5,7,8,11,12,13,15,17,18,20,21,26,36,37,38,39,40,41,42,43,44,45,46,47]. Concerning the research field of application of the WPHF protocols, 12 studies involved “extra force” [5,7,13,18,20,31,33,34,37,41,45,47], 9 “mechanisms” [8,11,12,15,17,35,36,38,40], 7 “fatigue” [21,26,39,42,43,44,46], and 4 “training” [16,29,30,32]. All studies were published between 2002 and 2021 (Table 1).

### 3.2. Risk of Bias within Studies

According to the EPHPP checklist of risk bias, the results of the quality assessment (Appendix A) showed 93.8% agreement between the reviewers. The score was moderate for 2 studies [5,41] and strong for the rest of the 32 studies, resulting in a low-bias risk quality of the studies included in the current review.

### 3.3. Participants

In total, 462 adults with 122 females (26.4%) were included in 32 studies. The mean age of the participants, adjusted for the number of participants in each study was 31.8 years. 309 participants received NMES on the TS (mean *n* = 16.3/study), and 115 on the TN (mean *n* = 11.5/study) and 39 on both TS and TN (mean *n* = 13.0/study). In 9 studies 22 dropouts out of 109 participants in total were reported (18.3% dropout rate). The dropout cases were 4 for studies including TS protocols, 13 for TN protocols and 5 for both. The most common reported reason for dropout was pain, with 12 cases (54.5%). 10 of these cases involved protocols with TN and 2 protocols with TS and TN.

### 3.4. Protocols

Most of the studies incorporated multiple protocols, resulting in total 87 NMES protocols, among which 43 were applied on the TS and 44 on the TN. All analyzed NMES parameters of these protocols are shown in Table 2. Figure 2 shows the distribution of each of the stimulus parameter expressed as a percentage of total TS or TN protocols.

### 3.5. Stimulation Frequency

Regarding the stimulation frequency (Figure 2A), 3 patterns of stimulation frequencies were identified: (a) the constant frequency pattern, with a continuous stimulation frequency, (b) the burst pattern, with a variable stimulation frequency following a low/high/low frequency scheme, and (c) the progressive pattern with a gradually increasing and decreasing frequency (ramp) profile. For the constant frequency, 26 protocols with frequencies ranging from 20 to 200 Hz were found for TS, and 33 protocols with frequencies ranging from 5 to 100 Hz were counted for TN. The burst pattern protocols were 16 for the TS, with 4 different modes (20/80/20, 20/100/20, 25/100/25 and 100/30) and 11 for the TN, with one mode (20/100/20). One progressive protocol was reported for TS, with a linearly increasing frequency from 4 to 100 Hz and decreasing from 100 to 4 Hz within 6 s. No progressing stimulation protocol was reported for TN.

### 3.6. Train Duration

For the constant frequency protocols, the start and end of stimulation expressed the duration of the (single) train. In both TS and TN, the most common duration was >10–20 s (14 for TS and 19 for TN), followed by the >5–10 s (6 for TS and 11 for TN, Figure 2B). For the burst protocols the duration of the burst ranged from 0.25 to 16 s, whereas the duration of the stimulation before and after the burst ranged from 2 to 3 s.

### 3.7. Stimulation Intensity

The current intensity of the stimulus was reported in 19 of the 32 studies in terms of mA (8 as range and 11 as mean ± standard deviation). For all protocols intensity was reported as the percentage of produced force relative to the maximal voluntary contraction (41 for TS and 16 for TN, with values ranging from 2 to 40% of MVC) or the recorded M-wave expressed as percentage of the maximum M-wave (M_max_; none for TS and 13 for TN, with values ranging from 0.3 to 20%) or as a percentage of the motor threshold (MT; none for TS and 15 for TN, with values ranging from below MT to 1.5 times MT). In one study the intensity of the TS protocol was adjusted to each subject’s tolerance level [32], and in one study with TN protocol, the intensity was variable, and progressively adjusted to maintain 20% of MVC [21].

### 3.8. Pulse Width

The majority of TS and TN protocols had 1 ms pulse width (38 and 40, respectively; Figure 2D). Three TS and 2 TN protocols had 0.5 ms pulse width. Furthermore, 2 TS [15,32] and 2 TN protocols [11] had shorter than 0.5 ms pulse duration.

## 4. Discussion

Eighty-seven protocols targeting the afferent pathway were identified and retrieved from 32 studies that fulfilled the inclusion criteria of this review. The number of studies was greater for the TS than TN, but a greater number of protocols with greater diversity in their characteristics was observed in TN. It was found that TN had a wider range of stimulation frequencies and methods to define the intensity of the stimulation. Furthermore, pain could be a limiting factor for the application of TN stimulation, with similar characteristics as in TS.

Regarding the pulse width, the most frequently used duration was 1 ms, whereas the pulse width of 0.5 ms or lower was used in fewer protocols for both stimulation types. The stimulation site (TS or TN) seems to have no influence on the force levels when 1 ms pulse width is used at 100 Hz [7]. Furthermore, although wider pulses of stimulation (i.e., 1 ms) seem to be less metabolically demanding than short-pulse NMES [47], and are used to stimulate more preferably the afferent pathway [8,9,10], there is evidence for TS that NMES with briefer pulse width (0.26 or 0.5 ms) may induce similar functional benefits (performance in mobility tests) after NMES training [32] or similar torque decline (volitional and electrically-induced) after an NMES fatigue protocol [44]. There are also indications that short (0.2 ms) or wide (1 ms) pulse width induce a similar afferent output over the contralateral upper limb when stimulating the biceps brachii at 10% of MVC [48]. Furthermore, twitch torque potentiation was the same between short or wide pulse stimulation [45]. Therefore, short and wide pulse durations might have similar functional effects during muscle stimulation. It has to be noted though that at higher stimulation intensities, even with wide-pulse NMES, the antidromic efferent volley may cancel out the reflexively-induced muscle activation [8].

For the TS protocols, the most prominent frequency was the constant-frequency of 100 Hz with fewer counts at other frequencies. For TN, 100 Hz was again the most prominent frequency but more protocols than TS had frequencies between 5 and 80 Hz. There is evidence that corticospinal excitability is affected after NMES training using low (20 Hz) or high (100 Hz) stimulation frequencies, whereas spinal excitability was only changed after high-frequency NMES [30]. This is in line with experiments examining extra force using NMES protocols with different stimulation frequencies, showing that reflex excitability decreased after NMES at 20 Hz, did not change at 50 Hz and increased at 100 Hz [13]. Therefore, it seems that the stimulation frequency, at least for the TN, influences the nature of spinal adaptations. However, further work using similar stimulation protocol is needed to provide a clear conclusion on this aspect.

In the cases of high frequency stimulation bursts, the most common condition was 20/100/20 Hz for both TS and TN. Minor variations of using 25 Hz instead of 20 Hz before and after the burst have assumably negligible effects, although no studies have examined explicitly such comparisons. In contrast to constant frequency, burst stimulation occurs when the motor axons are already active, and this activation is enhanced after the burst (extra force), due to both muscular [43] and neural contributors [18]. It is worth noting however, that a 2-s burst at 100 Hz can augment the central contribution at similar level, independent of the stimulation site [17]. Therefore, repetitive short bursts which can be better tolerated than continuous nerve stimulation, could be a useful alternative to examine and restore spinal and supraspinal functions.

Regarding the intensity of stimulation all TS protocols used a force level using MVC as a reference point, whereas TN protocols used in addition to this, the amplitude of M_max_ and motor threshold as well. The vast majority of protocols, both for TS and TN, induced stimulation currents that produced up to 10% of MVC. TN protocols with intensities set up to 5% of M_max_ or 1.1–1.5 times above the motor threshold correspond within this MVC range (i.e., ~10%; 38). However, since electromyogram, required for measuring M_max_, is not necessarily measured in studies with TS, it could be suggested to report in future studies with TN the produced force expressed as percentage of MVC, in order to have a common ground regarding the intensities used in protocols with TS. Additionally, studies that reported intensities above 10% of MVC aimed to investigate more functionally relevant contractions, imitating produced torques at ranges required during walking or standing [17]. Besides, low level of initial force might minimize antidromic collision on motor axons and facilitate the potential input from the spinal cord to the muscles [35,37]. This is also in line with the fact that contractions at low levels of MVC may enable the development of extra force, which is a useful tool to investigate central and peripheral contributions [8].

It is important to mention that in the current review discomfort was reported as a cause of dropout only in studies with TN or with both TN and TS protocols. In the study of Neyroud et al. [7], the current intensity required to induce force at 5% of MVC, as well as the discomfort level, was lower in TS compared with TN. Hence, it could be argued that higher MVC levels could result in more discomfort during TN stimulation and could explain the reduced number of protocols above 10% of MVC. It is characteristic that in the study of Martin et al. [26] intensity was set to result in a force at 20% of MVC and discomfort rate was the greatest among all included studies. Consequently, the rising question is whether TS NMES has similar effects compared to TN NMES. From the three studies that compared the two approaches, Neyroud et al. [7] reported that extra forces have similar magnitude independent of the stimulation site, whereas Baldwin et al. [18] demonstrated greater extra forces when the muscle was stimulated. On the other hand, it seems that muscle and nerve stimulation involve different pathways [18] and that TN NMES may have the potential to involve more sensory pathways than TS [17]. It is also worth noting that according to findings of the analyzed articles, the involvement of the sensory pathway might be facilitated in the presence of voluntary contraction [35], depressed by transcutaneous electrical nerve stimulation [31], and depressed by antagonist muscle contraction [37]. These latter results raise the opportunity to develop strategies that may amplify or depress the central contribution to force production during NMES, depending on the objective of the treatment.

Finally, among the included studies there might be several methodological sources of variability that could influence the outcome variables or even the final conclusions. None of the studies reported whether their participants were restricted from strenuous activity or alcohol/coffee consumption prior to testing. Furthermore, different size of stimulating electrodes between studies and variations in electrode positioning might be another source of variability in the effectiveness of NMES. It is worth noting that electrodes of variable length (presumably adapted to the participants’ size) have been used in only one study [5]. Regarding the position of the electrodes, all studies used fixed measures from anatomical reference points. However, electrode placement over the motor point may increase the efficiency of NMES, with minimal injected current and less discomfort [49]. Since this might vary between subjects [50], motor point detection for each individual is recommended prior electrode placement. All these should be considered in future studies for more reproducible and robust results that can elucidate the physiological mechanisms underlying NMES.

## 5. Conclusions

This review describes the properties of protocols that applied TS and TN stimulation, targeting to the involvement of the central nervous system through afferent pathways. After analyzing 32 studies and 87 protocols the most common practice to assess the afferent pathway with NMES on TS or TN is with 100 Hz frequency, 1 ms pulse duration, and stimulation intensity at 10% of MVC. However, lower frequencies and pulse widths may also be effective and should be considered, especially in cases that discomfort due to the stimulation is an issue. This variety of NMES protocols that can induce activation a muscle via spinal pathways, dictates the need for further research to pinout the potential differences between different protocols and to create tailored protocols depending on the application and the goal of the research question, with the ultimate goal to optimize their effectiveness.

## Figures and Tables

**Figure 1 sensors-23-02347-f001:**
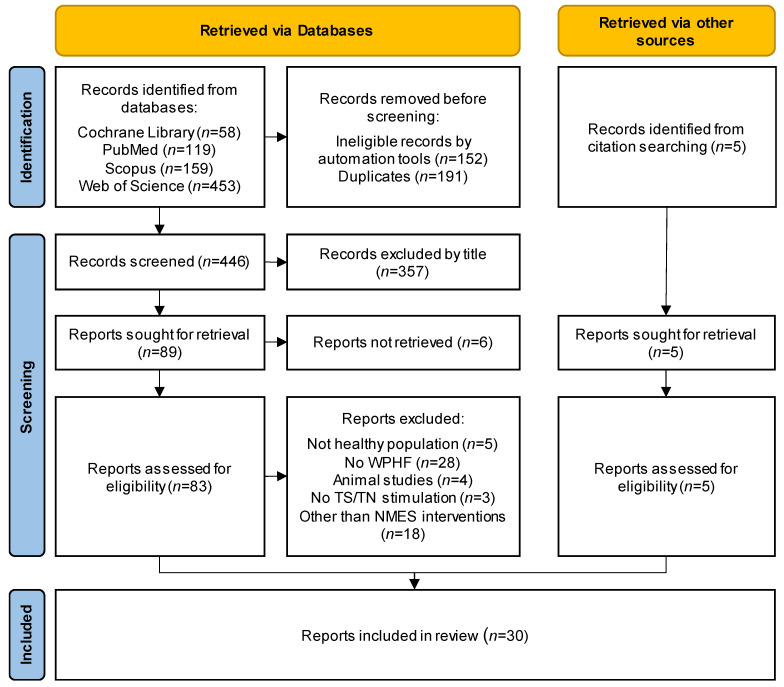
Flow diagram of the study selection process via databases and other sources, according to PRISMA 2020 for new systematic reviews.

**Figure 2 sensors-23-02347-f002:**
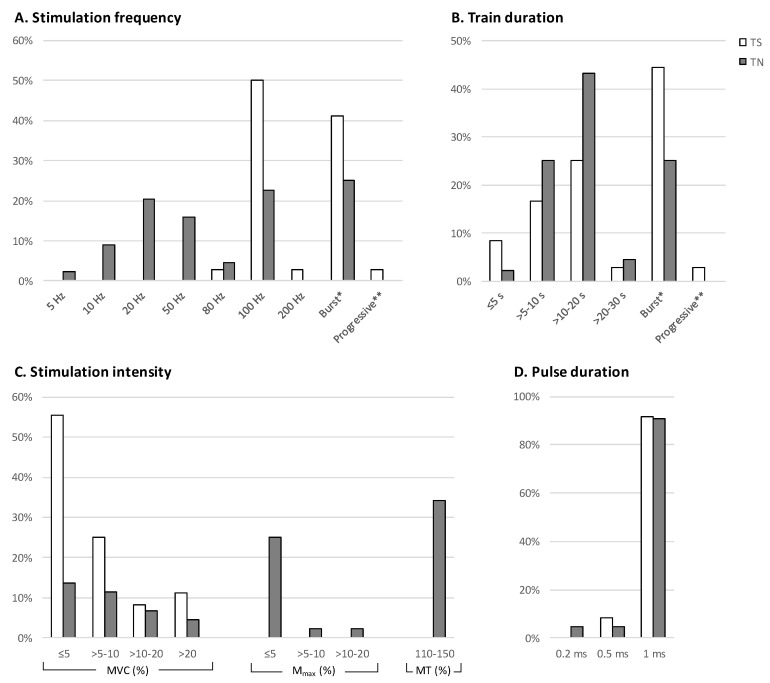
Incidence of stimulation frequency (**A**), train duration (**B**), stimulation intensity (**C**) and pulse duration (**D**) expressed as percentage of total number of WPHF protocols applied on the triceps surae (TS) and tibial nerve (TN). * Burst represents protocols with variable frequency comprised of one WPHF burst (80–100 Hz) between stimulation trains of lower frequency (20–30 Hz). ** Progressive represents one protocol with variable stimulation frequency ramping up from 4 to 100 Hz and ramping down to 4 Hz. Stimulation intensity (C) has been adjusted based on the produced force [expressed as a percentage of maximal voluntary contraction (MVC)] or the M-wave [expressed as percentage of maximum M-wave (M_max_)], or a percentage of the motor threshold (MT).

**Table 1 sensors-23-02347-t001:** Characteristics, scope, and main conclusions of all included studies with WPHF NMES on the triceps surae (TS), tibial nerve (TN), or both.

First Author, Year (Reference Number)	Study Design *Research Field*	n/Females	Age (Years)	Dropouts/Discomfort (Rate)	Subject/Scope of the Study	Main Conclusions
**Triceps surae NMES studies (*n* = 19)**						
Collins, 2002 [5]	Cohort *Extra force*	21/7	-	-	To investigate the optimal stimulus properties for evoking plateau-like phenomena to motoneurons	Experimental proof of contractions generated by central contribution with activation of plateau-like potentials.
Dean, 2007 [8]	Cohort *Mechanisms*	8/2	21–43	-	The effect of NMES parameters (frequency, duration, and intensity) on the central contribution to torque using several stimulation patterns	Torque was generated via stimulation of motor axons at low-frequency (≤20 Hz) NMES and via stimulation of sensory axons at high-frequency (≥80 Hz), low intensity, and wide-pulse duration NMES.
Dean, 2008 [37]	Cohort *Extra force*	10/1	18–42	-	Methods to activate the antagonist muscles, with practical implications for functional electrical stimulation.	Plantar flexor extra torque can be diminished by volitional or electrical activation of the antagonist muscles
Lagerquist, 2009 [41]	Cohort *Extra force*	13/0	28–53	-	To enhance the central contribution to the evoked contractions by using stimulation frequencies of 20 Hz and 100 Hz.	During NMES evoked torque increases possibly due to the contribution of central neural pathways, since blocking the antidromic volley the same NMES protocol demonstrates a decrease in evoked torque.
Neyroud, 2014 [42]	Cohort *Fatigue*	14/3	27.0 ± 4.0	-	Investigation whether central recruitment occurs during WPHF contractions and comparison of the extent and origin of muscle fatigue	WPHF fatigue protocol induced fatigue at faster rate than a fatigue protocol with short pulse duration and low frequency.
Papaiordanidou, 2014 [43]	Cohort *Fatigue*	10/0	32.0 ± 3.8	-	To understand the nature of fatigue induced by two NMES protocols, matched for the number of delivered pulses, and to determine whether the stimulation pattern induces different neuromuscular adaptations.	Variable frequency fatigue protocol (with doublets at 100 Hz) demonstrated greater torque decrease than constant frequency, whereas both protocols had similar effects on all examined muscular, spinal and supraspinal mechanisms.
Papaiordanidou, 2014 [44]	Cohort *Fatigue*	8/-	27.8 ± 7.1	-	Examination of fatigue development during NMES protocols with different stimulation frequencies and pulse widths.	Evoked torque decreased at similar extent relative to the initial value, independent of the pulse duration.
Regina Dias Da Silva, 2015 [45]	Cohort *Extra force*	13/4	30.0 ± 7.0	-	Examination whether the twitch potentiation would be greater following conventional than WPHF or voluntary contractions of the plantar flexor muscles	There is no different in twitch torque potentiation when using WPHF or short-pulse low-frequency NMES.
Wegrzyk, 2015 [34]	Cohort *Extra force*	42/22	28.0 ± 6.0	-	Evaluation of twitch potentiation, H-reflex, and M-wave to understand the neuromuscular mechanisms of extra forces in response to WPHF.	Individuals that exhibit extra force demonstrate depressed H-reflex after NMES, which suggests the contribution of central mechanisms.
Wegrzyk, 2015 [47]	Cohort analytic *Extra force*	18/5	29.0 ± 7.0	-	Comparison of metabolic demand during WPHF NMES and voluntary contractions, using ^31^P-magnetic resonance spectroscopy.	Extra force induced by WPHF is less demanding metabolically than short-pulse low-frequency NMES, and possibly exhibits a muscle activation pattern similar to voluntary contractions.
Cheng, 2017 [36]	Cohort *Mechanisms*	11/5	30.0 ± 7.0	-	Investigation of the mechanisms underlying the low-frequency force potentiation following high-frequency stimulation.	High-frequency wide-pulse NMES results in a potentiation in torque produced by a subsequent low-frequency NMES train possibly due to increased sensitivity of myofibrillar Ca^2+^.
Wegrzyk (2017) [15]	Cohort *Mechanisms*	18/6	26.0 ± 5.0	2/0(11.1%/0%)	Investigation of the cerebral activation pattern during WPHF NMES protocols, as compared to voluntary contractions, matched for the same initial isometric force.	NMES-induced isometric contractions resulted in brain activation patterns including sensorimotor areas and subcortical structures, similar to the activation patterns of voluntary movements.
Grosprêtre, 2017 [39]	Cohort *Fatigue*	10/3	24.6 ± 4.2	-	Acute fatigue effects of submaximal NMES delivered at different frequencies during NMES-evoked and maximal voluntary contractions.	For the same exerted torque, high frequency/low intensity fatigue protocols induce greater fatigue, whereas low or high frequency stimulation induce alterations at muscular or neural (spinal and supraspinal) level, respectively.
Grosprêtre, 2018 [40]	Cohort *Mechanisms*	10/3	24.6 ± 4.2	-	To understand the origins of spinal excitability modulation after NMES.	Spinal excitability decreased and presynaptic inhibition increased after NMES, especially at high-frequency mode.
Mani, 2018 [32]	RCT *Training*	30/17	73.5 ± 4.8	2/0 (6.6%/0%)	To compare the effect of a 6-week NMES protocol with long and short pulse duration on the mobility of older adults	Wide- and short-pulse NMES training resulted in gains in functional mobility tests (walking speed, chair rise etc.).
Neyroud, 2019 [16]	RCT *Training*	10/4	24.0 ± 1.0	-	Evaluation of neuromuscular adaptations after 3 weeks of WPHF NMES	Evoked force time integral was increased after 3 weeks of WPHF NMES, with no changes in plantar flexor neuromuscular properties.
Bouguetoch, 2021 [29]	RCT *Training*	10/3	24.0 ± 5.8	-	Investigation of strength gains as well as the muscular and neural plasticity after training that combines motor imagery with NMES.	MVC peak twitch torque and M-wave amplitude increased after NMES training, whereas no changes in muscle architecture or other neural variables (H-reflex, V-wave) were observed.
Donnelly, 2021 [31]	RCT *Extra force*	23/3	26.7 ± 2	-	Evaluation of torque evoked by WPHF using transcutaneous electrical nerve stimulation (TENS) or transcutaneous spinal direct current stimulation (tsDCS).	Extra force produced by WPHF stimulation was diminished after a bout of TENS but was not affected after a bout of tsDCS.
Espeit, 2021 [33]	RCT *Extra force*	30/5	26.6 ± 6	-	Evaluation of the extra force magnitude with high and low frequency, wide pulse stimulation.	Low and high frequency wide pulse NMES has similar number of responders for extra force, with higher extra force for high frequency.
**Tibial nerve NMES studies (*n* = 10)**						
Klakowicy, 2006 [20]	Cohort *Extra force*	11/-	21–42	2/1 (18.1%/9.1%)	Examination of whether the extra torque generated by 100 Hz stimulation involves the activation of spinal motoneurons and whether H-reflexes can recover during tetanic stimulation of soleus afferents	Production of extra force was accompanied by an increase in the H-reflex amplitude with no change in the M-wave amplitude.
Lagerquist, 2010 [11]	Cohort *Mechanisms*	18/2	19–43	4/4 (22.2%/22.2%)	The effect of pulse width on M-wave, H-reflex, and torque during 2 s of 100 Hz stimulation	Wide-pulse NMES generated greater force via sensory pathways.
Clair, 2011 [35]	Cohort analytic *Mechanisms*	11/3	20–46	-	Investigation of H-reflex recovery after 10-s trains of stimulation at 5–20 Hz, during functional tasks and low-level muscle contractions.	Increasing the stimulation frequency of the afferent pathway and the intensity of background voluntary activation can recover the H-reflex amplitude, when it has been reduced due to post-activation depression.
Lagerquist, 2012 [12]	Cohort *Mechanisms*	10/3	22–44	-	Evaluation of the enhanced spinal and corticospinal excitability of the soleus muscle following voluntary plantar-flexions, NMES of the TN and combination of the two.	Spinal excitability increased after a contraction induced by NMES combined with voluntary activation, whereas there was no effect on corticospinal excitability.
Dean, 2014 [38]	Cohort *Mechanisms*	9/2	22–44	2/- (22.2%/-)	Characterization of the recruitment and ongoing discharge of human motoneurons when they receive trains of afferent impulses over a range of physiologically relevant frequencies	Repetitive stimulation of the TN, with intensity that does not elicit H-reflex or M-wave with a single pulse recruited motor units, and this was retained even after cessation of the stimulation.
Doix, 2014 [21]	Cohort *Fatigue*	9/0	26.6 ± 2.0	1/1 (11.1%/11.1%)	Investigation of muscle fatigue of the TS after NMES protocols with constant or increasing current intensity and comparison with fatigue induced by voluntary contractions.	MVC decreased, and neural properties changed to a similar extent independent of the variable or constant stimulation intensity fatigue protocol.
Martin, 2016 [26]	Cohort *Fatigue*	15/3	28.0 ± 8.0	4/4 (26.4%/26.7%)	Examination of whether the WPHF paradigm applied over the nerve trunk limits muscle fatigue by favoring motor unit’s recruitment through afferent pathways, in contrast to the motor unit activation through efferent pathways associated with a conventional NMES protocol	WPHF induced greater H-reflex increase and greater M-wave decrease compared to short pulse low frequency stimulation, despite no differences in force reduction.
Vitry, 2019 [13]	Cohort *Extra force*	12/2	27.1 ± 8.7	-	Examination of the conditions to invoke extra torque by modulating the frequency and intensity of stimulation.	Spinal excitability contribution to extra forces is frequency dependent, whereas supraspinal mechanisms do not seem to be affected by the frequency of NMES.
Vitry, 2019 [46]	Cohort *Fatigue*	9/2	23.2 ± 6.6	-	The effect of stimulation frequency on neuromuscular fatigue using stimulation parameters favoring an indirect motor unit recruitment through the afferent pathway.	Low and high frequency fatigue protocols induced similar level of decrease in twitch torque and level of voluntary activation.
Vitry, 2019 [30]	RCT *Training*	10/3	22.7 ± 6.4	-	The effect of chronic application of NMES training modalities using wide pulse duration, low stimulation intensity, and nerve stimulation to maximize the central contribution to the evoked torque	Similar strength gains independent of the stimulation frequency NMES training, whereas low frequency NMES resulted in supraspinal adaptations, and high frequency NMES resulted in supraspinal and spinal adaptations.
**Triceps surae and tibial nerve NMES studies (*n* = 3)**						
Baldwin, 2006 [18]	Cohort *Extra force*	15/2	20–41	1/1 (6.7%/6.7%)	The optimal technique to evoke sustained contractions enhanced through spinal pathways during NMES	Larger extra forces are produced with muscle stimulation in comparison to nerve stimulation, whereas stimulating the muscle or nerve may involve different neural pathways.
Bergquist, 2011 [17]	Cohort *Mechanisms*	14/2	20–48	4/1 (28.5%/7.1%)	To compare the contributions of central and peripheral pathways to the motor unit recruitment for contractions of similar amplitude generated by NMES applied over TN or TS.	NMES over the tibial nerve produced responses with greater involvement of the sensory pathway than stimulation over the triceps surae.
Neyroud, 2018 [7]	Cohort *Extra force*	10/5	28.0 ± 4.0	-	Evaluation of the reliability of force production induced by WPHF NMES delivered over the TN or the plantar flexor muscle.	Nerve and muscle NMES resulted in similar magnitude of extra forces.

Dashes (-) designate no available information. RCT: randomized controlled trial. The research field of each study is shown in the second column in italics.

**Table 2 sensors-23-02347-t002:** NMES protocols and stimulation parameters (pulse width, frequency, duration, and intensity).

First Author, Year (Reference Number)	NMES Protocols (*n*)	Neuromuscular Electrical Stimulation Parameters (NMES)			
		Pulse Width (ms)	Frequency (Hz) × Duration (s) × Count of Bursts within Cycle	Duty Cycle on/off × Count of Cycles	Intensity
Extra force (*n* = 12)					
Collins, 2002 [5]	TS: 4	1	C: (100, 200) × 7” B: 100 × 2” × 1 P: 100 × 6”	7”/0” × 1 6”/0” × 1	5% MVC 5% MVC 5% MVC
Baldwin, 2006 [18]	TS: 2 TN: 2	1	B: 100 × 2”	7”/3” × 5	2, 4% MVC
Klakowicy, 2006 [20]	TN: 1	1	B: 100 × 2”	7”/3” × 5	0.3–4% M_max_
Dean, 2008 [37]	TS: 1	1	B: 100 × 2” × 4		3% MVC
Lagerquist, 2009 [41]	TS: 3	1	C: 100 × 30” B: 100 × 2” × 4 C: 100 × 30”	1”/1” × 15	13.0 ± 2.7 mA 7.5% MVC
Regina Dias Da Silva, 2015 [45]	TS: 1	1	C: 100 × 10”		16.5 ± 10.3 mA 10% MVC
Wegrzyk, 2015 [34]	TS: 1	1	C: 100 × 20”	20”/90” × 5	12.9–15.4 mA ~5% MVC
Wegrzyk, 2015 [47]	TS: 1	1	C: 100	20”/20” × 20	15–18 mA 8.5–11% MVC
Neyroud, 2018 [7]	TS: 1 TN: 1	1	C: 100	20”/40” × 10	2.3–4.6 mA 5% MVC
Vitry, 2019 [13]	TN: 15	1	C: (20, 50, 100) × 20”		1.1, 1.2, 1.3, 1.4, 1.5 MT
Donnelly, 2021 [31]	TS: 1	1	C: 100	20”/40” × 1 or 3	5% MVC
Espeit, 2021 [33]	TS: 2	1	C: 20 Hz C: 100 Hz	20”/90” × 3 20”/90” × 3	9 ± 3 mA 8 ± 3 mA 10% MVC
Mechanisms (*n* = 9)					
Dean, 2007 [8]	TS: 8	1	C: 50 × 20” C: 100 × 20” B: 80 × 2” × 4 B: 100 × 2” × 4		1, 3% MVC 1, 3% MVC 3% MVC 1, 3, 5% MVC
Lagerquist, 2010 [11]	TN: 6	0.2, 0.5, 1	B: 100 × 2”	7”/n.r. × 4	5–15 mA1–2, 5% M_max_
Bergquist, 2011 [17]	TS: 2	1	B: 100 × 2”	8”/45” × 5	28.3 ± 1.9 mA 10% MVC 34.2 ± 2.7 mA 20–40% MVC
	TN: 4	1	C: 20 × 8” B: 100 × 2”	8”/45” × 5	7.8 ± 0.9 mA 10% MVC 8.4 ± 0.8 mA 20–40% MVC
Clair, 2011 [35]	TN: 6	1	C: (1,5,10,20) × 10”	10”/30” × 3	1,5,10,20% M_max_
Lagerquist, 2012 [12]	TN: 1	1	C: 100 × 40′	5”/5” × 240	2–3% MVC
Dean, 2014 [38]	TN: 2	1	C: (80, 100) × 30”		2.5–10 mA ≤1 MT
Wegrzyk, 2017 [15]	TS: 2	0.051	C: 25 C: 100	20”/20” × 20 20”/20” × 20	135 ± 31 mA 25.0 ± 11.0 mA 10% MVC
Cheng, 2017 [36]	TS: 1		B: 100 × 2” × 1	6”/0” × 1	17.0 ± 11.0 mA 5–10% MVC
Grosprêtre, 2018 [40]	TS: 1	1	C: 100	6”/6” × 40	12.8 ± 4.3 mA 20% MVC
Fatigue (*n* = 7)					
Doix, 2014 [21]	TN: 2	1	C: 50 × 6” C: 50 × 6”	6”/6” × 2 6”/6” × 2	20.6 ± 1.7 mA variable20% MVC
Neyroud, 2014 [42]	TS: 1	1	C: 100	20”/40” × 20	18.0 ± 8.0 mA 10% MVC
Papaiordanidou, 2014 [43]	TS: 1	0.5	C: 30 B: 100 doublets	0.167”/0.5” × 450 0.146”/0.5” × 450	30% MVC
Papaiordanidou, 2014 [44]	TS: 2	0.5, 1	C: 100	4”/6” × 60	30% MVC
Martin, 2016 [26]	TN: 1	1	C: 80	6”/6” × 40	20% MVC
Grosprêtre, 2017 [39]	TS: 1	1	C: 100	6”/6” × 40	12.9 ± 1.4 mA 20% MVC
Vitry, 2019 [46]	TN: 2	1	C: 20 C: 100	20”/20” × 25 20”/20” × 25	11.3 ± 5.6 mA 11.9 ± 6.0 mA 10% MVC
Training (*n* = 4)					
Mani, 2018 [32]	TS: 2	0.261	C: 50C: 100	4”/12” × 75 4”/12” × 75	33 ± 10 mA 18.5 ± 8 mA until tolerance level
Neyroud, 2019 [16]	TS: 1	1	C: 100	20”/40” × 10	13.3 ± 9.1 mA 5% MVC
Vitry, 2019 [30]	TN: 2	1	C: 20 C: 100	20”/20” × 25 20”/20” × 25	12.2 ± 6.4 mA 11.2 ± 4.7 mA 10% MVC
Bouguetoch, 2021 [29]	TN: 1	0.5	C: 80	6”/6” × 40	10–45 mA 20% MVC

**Abbreviations:** TS: triceps surae; TN: tibial nerve; C: constant frequency; B: variable frequency with a burst of higher frequency within each cycle (numbers correspond to the frequency and duration of the burst); P: progressively increasing and decreasing frequency (numbers correspond to the peak frequency and duration or the ramp-up and ramp-down); MVC: maximal voluntary contraction; MT: motor threshold; M_max_: maximum M-wave; n.r.: not reported.

## Data Availability

Not applicable.

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
