# Peer review of "Protocols Targeting Afferent Pathways via Neuromuscular Electrical Stimulation for the Plantar Flexors: A Systematic Review"

_sensors, 2023, doi:10.3390/s23042347_

Round 1
Reviewer 1 Report
This review describes NMES protocols that provide evidence for synaptic stimulation of motor neurones. Is focuses on stimulation of the triceps surae muscles (TS) and the tibial nerve (TN).
Line 41-42: Change to "...has the potential to preferentially activate motor neurons via spinal pathways."
Line 77: Change to "...stimulation frequency greater than or equal to 80 Hz,.."
Line 87: Reference is made to Figure 1, which I was unable to find in the manuscript file and the supplementary materials file which included Appendices 1 and 2 only.
Figures 2A (Line 153), 2B (Line 164) and 2D (Line 175) are also mentioned but I cannot find the figures.
The Discussion is well written and comprehensive.
Author Response
Please, see attachment.

Reviewer 2 Report
Comments to Authors
One might say that NMES has been used since the early 18th century, since the invention of the Leyden Jar but if we limit its history to the time when portable electronic stimulators could produce accurate current pulses (say 1960), practitioners used to expect that the neuromuscular responses were only due to excitation of motor nerve fibres. In the past 20 years, there has been a realisation that there can also be motor output due to activation of the afferents. I am not involved with this research but it appears to me that there is a disagreement between published studies. On one hand, ref 9 clearly states and shows that there are action potentials of three types: direct motor responses (M-wave), H-waves and also those that are not time-locked to the stimulus pulses. The later can contribute a large component of the total force and continue long after stimulation has ceased. On the other hand, ref 26 reports M and H but no non-time-locked action potentials. Since these are both studies in able-bodied people, the disagreement is surprising and it is important to find out what the difference is between the experiments. Indeed, it would be very interesting to know which of all the studies cited here are in the first (including non time-locked) or second (only time-locked) responses.
However, your paper does not present this information. Most reviews compare the results from many studies but yours compares the methods. It tabulates the participant information, their age, dropouts, pain, and stimulus intensity, pulse width and frequency, but not the extent to which afferent responses occurred. At line 66, you say “this review aims to document and describe the attributes of NMES protocols that give evidence of stimulating the motor neurones synaptically …” which is true, but not to include the results, having located all the relevant papers, is disappointing. I wonder whether you intended to include results but they are too varied to categorise.
So, I see this as a useful paper, guiding interested readers to the relevant literature. Also it is interesting that the range of pulse parameters is so narrow, but perhaps the most urgent question is whether similar results can be obtained in different labs with the same parameters. What can the differences be? Is it to do with some uncontrolled variable such as exact electrode position, electrode area, whether biphasic pulses are in use, or whether the participants had drunk coffee in the previous hour?
Minor point: there seems to be no Figure 1.
Author Response
Please, see attachment.

Reviewer 3 Report
The study it is well planned and executed. The stuy is relevant affecting a branch less explored of FES. The PRISMA protocol is well applied.
The contribution is confusing and hard to follow. The division in four types of studies has no other consequences on the results, and therefore is irrelevant or has to be included in the results. For instance, do type of study influences the protocol?
The discussion should be more centered on the results. Some sentences look contradictory (or are in fact contradictory) but not position is set in the contradiction. For instances in lines 186-187 the authors declare that "it is well known that wider pulses of stimulation can stimulate more preferably the afferent pathway, NMES on the TS with different pulse width 187 durations (0.5 and 1 ms) did not affect significantly the torque decline during NMES" how can this be possible? The "well known" knowledge is wrong or there are other mechanisms playing a role? What did authors suggest?.
From the other hand. Do authors considered other way of arranging information in tables rather than author based? Seems that would be more clear the tables arranged protocol-based and then summing up the articles in each of the protocols.
Author Response
Please, see attachment.

Reviewer 4 Report
The study by Papavasileiou and colleagues used a systematic review to analyze the neuromuscular electrical stimulation protocols (NMES) on the plantar flexors (through triceps surae (TS) or tibial nerve (TN) stimulation) to stimulate afferent pathways.
1. I used PRISMA 2020 checklist to review the submission, and some parts were able to identify each item. Some items I could not identify, and I suggest the authors go through the checklist and ensure all items are addressed, and if "not relevant", indicate such in their narrative. Please add the updated PRISMA 2020 checklist and check your manuscript.
2. Eligible criteria, the orginal research articles should be addressed specially (e.g., RCT, or Cohort study...). The participants can be described as healthy adults. Additionally, please describe your study outcome in the eligible criteria according to the PICO principles.
3. Although this manuscript offered some strength, what's the limitations of your systematic review? Authors should list the limitations to provide guidance to further research.
Author Response
We thank the Reviewer for his/her time reading our manuscript and exposing the essence of our work.
Round 2
Reviewer 1 Report
The manuscript is much improved and, in my opinion, now meets the standard for publication.
Reviewer 4 Report
The authors have satisfactorily addressed my concerns. I do not have any more questions.